# Modular Digital and 3D-Printed Dental Models with Applicability in Dental Education

**DOI:** 10.3390/medicina59010116

**Published:** 2023-01-06

**Authors:** Alexandru Eugen Petre, Mihaela Pantea, Sergiu Drafta, Marina Imre, Ana Maria Cristina Țâncu, Eduard M. Liciu, Andreea Cristiana Didilescu, Silviu Mirel Pițuru

**Affiliations:** 1Department of Prosthodontics, Faculty of Dentistry, “Carol Davila” University of Medicine and Pharmacy, 17–23 Calea Plevnei, 010221 Bucharest, Romania; 2Coordinator of the 3D Printing Department, Center for Innovation and e-Health (CieH), “Carol Davila” University of Medicine and Pharmacy, 20 Pitar Mos Str., 010454 Bucharest, Romania; 3Department of Embryology, Faculty of Dentistry, “Carol Davila” University of Medicine and Pharmacy, 8 Eroii Sanitari Boulevard, 050474 Bucharest, Romania; 4Department of Professional Organization and Medical Legislation-Malpractice, Faculty of Dentistry, “Carol Davila” University of Medicine and Pharmacy, 020021 Bucharest, Romania

**Keywords:** dental students, digital dental models, digital technology, 3D printing, digital libraries, dental education, digital learning

## Abstract

*Background and Objectives*: The ever more complex modern dental education requires permanent adaptation to expanding medical knowledge and new advancements in digital technologies as well as intensification of interdisciplinary collaboration. Our study presents a newly developed computerized method allowing virtual case simulation on modular digital dental models and 3D-printing of the obtained digital models; additionally, undergraduate dental students’ opinion on the advanced method is investigated in this paper. *Materials and Methods*: Based on the digitalization of didactic dental models, the proposed method generates modular digital dental models that can be easily converted into different types of partial edentulism scenarios, thus allowing the development of a digital library. Three-dimensionally printed simulated dental models can subsequently be manufactured based on the previously obtained digital models. The opinion of a group of undergraduate dental students (*n* = 205) on the proposed method was assessed via a questionnaire, administered as a Google form, sent via email. *Results*: The modular digital models allow students to perform repeated virtual simulations of any possible partial edentulism cases, to project 3D virtual treatment plans and to observe the subtle differences between diverse teeth preparations; the resulting 3D-printed models could be used in students’ practical training. The proposed method received positive feedback from the undergraduate students. *Conclusions*: The advanced method is adequate for dental students’ training, enabling the gradual design of modular digital dental models with partial edentulism, from simple to complex cases, and the hands-on training on corresponding 3D-printed dental models.

## 1. Introduction

The digital revolution has spread across all medical fields, including dentistry [1], and, as expected, it has also touched the under- and postgraduate medical education. In dentistry, we are witnessing extraordinary progress generated by digital technology, such as clinical and technical procedures related to intraoral or extraoral scanning [1,2,3,4,5], 3D printing [6,7], guided dental implantology, complex maxillo-facial guided surgery or orthodontic digital planning [8], design and production of diverse prosthetic restorations using subtractive or additive methods [9,10,11,12], computerized validation of diverse fixed or mobile prostheses and computerized patient monitoring [1]. The importance of these beneficial elements constantly increases due to the encouraging perspectives in the development of digital dentistry. Today’s top trending elements in dentistry digitalization and in influencing the directions of scientific research in this field are considered to be the following: rapid prototyping (RP); augmented and virtual reality (AR/VR), artificial intelligence (AI) and machine learning (ML); personalized and precision medicine; and tele-medicine [13,14,15,16,17].

Dental students’ training holds a special place in this technological universe and, as a consequence, it must adapt to the characteristic rhythm of development of the actual digital era [18,19,20]. Thus, the concept of “virtual teaching and learning” (“e-teaching and e-learning”), which includes the use of digital dental didactic models, has been proposed and successfully applied in university education, facilitating students’ understanding of theoretical and practical aspects [21,22,23]; these methods ensure students’ efficient transition from a preclinical to a clinical context, thus proving their high educational value [21,22,23]. Haptic models employing virtual reality techniques represent an alternative to the standard models used in the preclinical training of students [24,25,26]. The famous Simodont (Nissin Dental Products Europe BV, Nieuw-Vennep, The Netherlands) was the first simulator that offered the possibility of learning proprioceptive abilities. The results of certain studies demonstrate that a clinically relevant qualitative feedback can be provided via a VR dental simulator [27,28]. While the cost of virtual simulation equipment is, unfortunately, still high, Nassar [29] pointed out in a relatively recent review that “none can deny the benefits of using these simulations strategies especially in saving faculty time as well as allowing the students to perform repeated attempts to achieve mastery at their own pace and after hours”.

On the other hand, digital applications are not intended to completely replace the traditional, conventional methods of teaching and learning [18,30], which have proved their efficiency and have already been validated. In this respect, the training on mechanical patient simulators (“phantom heads”) using standard dental models undisputedly remains an important stage in the preclinical training of dental students. Practical exercises performed on standard dental didactic models aim to form and, subsequently, to enhance the practical abilities of under- and postgraduate dental students, so that, in contact with real patients, they can demonstrate appropriate precision, correctness and expeditiousness in performing practical daily dentistry [31,32,33]. The dental models obtained with 3D printing have been used in dental education for some time. Deployment of 3D scanners, of CBCT and of 3D printers has enabled the precise replication of real anatomical structures, achieving simulated dental models having a real clinical correspondent, which can be successfully used in dental education [34,35] The combination of traditional simulation techniques with digital simulation broadens the educational possibilities towards decision-making training and deliberate practice, which are less possible in traditional settings.

Given the previously presented context, this paper has two major objectives: (Objective 1) to advance a new computerized method that allows the simulation of different virtual partial edentulism cases on modular digital dental models and the 3D printing of the obtained digital models; (Objective 2) to assess the opinion of a group of undergraduate dental students (*n* = 205) on the use of the proposed method in university dental education.

## 2. Materials and Methods

As previously mentioned, our study includes two distinct sections: the first one addresses a computerized method for obtaining modular digital and 3D-printed dental models and the second one presents undergraduate dental students’ opinion on the advanced method.

### 2.1. Obtaining the Modular Digital and 3D-Printed Dental Models

The construction of the simulation models is based on CAD/CAM technology and consist of the following phases:

Section 2.1.1: scanning of standard dental models (with intact anatomic teeth), and separate scanning of intact artificial teeth and of prepared artificial teeth for different types of fixed prosthetic restorations;

Section 2.1.2: based on the acquired data, simulated modular dental models are developed using CAD software;

Section 2.1.3: based on the previously obtained digital models, 3D-printed simulated dental models can be manufactured.

#### 2.1.1. Digital Data Acquisition

The proposed method for creating and accessing modular digital dental models is as follows. As a first step, digital dental models were achieved by digitalizing a maxillary and a mandibular standard dental model, their artificial teeth having intact anatomic dental crowns (Frasaco AN3, Frasaco GmbH, Tettnang, Germany). Subsequently, the upper and lower digital dental models were positioned in maximum intercuspation and scanned accordingly. All digital data corresponding to the dental arches, alveolar processes and maximum intercuspation position were obtained as .stl files (Standard Tessellation Language) using a Trios 3 intraoral scanner (3Shape, Copenhagen, Denmark). Successively, two groups of teeth were scanned separately, as follows: group A: represented by 32 artificial unprepared teeth, with intact anatomic dental crowns and their radicular extensions, corresponding to the full upper and lower dental arches, as produced by the manufacturer (Frasaco GmbH, Tettnang, Germany); group B: represented by 69 artificial teeth prepared for various types of fixed prosthetic restorations for total or partial coverage (Frasaco GmbH, Tettnang, Germany). These two groups were formed so that each unprepared artificial tooth, with an intact dental crown (from group A), corresponds to at least two prepared artificial teeth from Group B.

#### 2.1.2. Design of the Modular Dental Models

The digital models of the two dental arches were uploaded in a prosthetic restorations design application (CAD) (Ceramill Mind, Amann Girrbach AG, Koblach, Austria) and, observing anthropometric criteria, were mounted in the virtual articulator (Figure 1).

The .stl files corresponding to the intact (unprepared) artificial teeth that were separately scanned (group A) were uploaded one by one, in the same order, in the form of “Generic Visualization Mesh”, by indexing them with the intact dental models. The scanned images were indexed by reference to the corresponding surfaces of the dental crowns (Figure 2). The same procedure was applied for all the teeth of the two dental arches. Each artificial tooth scanned image (from Group A) was exported as a standard .stl file, maintaining the reference system.

The scanned images of the prepared artificial teeth belonging to group B were uploaded one at a time, also in the form of “Generic Visualization Mesh”, by indexation in relation with intact teeth models, using as reference the corresponding surfaces of the radicular extensions (Figure 3). The same procedure was applied for all prepared teeth from group B. Each prepared scanned image was exported as a standard .stl file, observing the reference system. The areas of the digital dental model corresponding to alveolar processes were isolated using the “Edit Mesh” tool (teeth are separated and their corresponding alveolar spaces are “closed”) (Figure 4). The two dental alveolar processes, corresponding to upper and lower dental arches, were exported as .stl files, observing the reference system.

Following the steps previously presented, the system of modular didactic digital dental models was created. This system is formed of: 1. edentulous ridge models; 2. unprepared teeth models (group A); 3. prepared teeth models (group B). It is important that all models should observe the same reference system so that, when combined, they will be placed in the correct position in relation to both their adjacent teeth and their antagonists. The resulting modular digital models allow students to perform repeated virtual simulation of any possible partial edentulism scenario. In order to generate a didactic modular digital dental model corresponding to a certain configuration, a new order is created in the CAD (Computer Aided Design) program. The two dental alveolar processes are uploaded as a digital wax-up, to which Boolean additions of prepared or unprepared teeth are performed.

We present a clinical situation corresponding to the absence of tooth #2.6/first upper left molar (edentation of 2.6), where the teeth next to the edentation (the abutment teeth) are prepared for complete fixed prosthetic coverage, with a circular cervical shoulder. At the lower arch, we generate a digital dental model corresponding to the edentation of all molars. In order to obtain the complete system of the modular didactic digital dental model, the following steps are to be followed: the .stl files corresponding to the two alveolar processes, respectively “MX GINGIVA FILLED.stl” (maxillary alveolar process) and “MD GINGIVA FILLED.stl” (mandibular alveolar process) are uploaded as digital wax-ups; the .stl files of the unprepared teeth from #1.8 to #2.4 and tooth #2.8 (Figure 5), and the respective files of the prepared teeth #25 and #27—the abutments—(Figure 6) are added to the “MX GINGIVA FILLED.stl” file as Boolean addition; the .stl files of the unprepared teeth from #3.5 to #4.5 are added to the file “MD GINGIVA FILLED.stl” (Figure 7). The areas corresponding to edentulous ridges can be edited in order to simulate bone resorption (Figure 8).

Moreover, this method can also use a dental model generated from a real patient. In this case, the following steps are required: 1. obtain digital intraoral impression; 2. generate models with removable dies/abutments; 3. print the removable dies/abutments; 4. prepare the abutments from the previous point (3) for different types of crowns; 5. re-scan the printed model with intact teeth and prepared teeth (abutments). A digital library is thus created, as in the case of the commercial didactic model.

All procedures in this scientific study were conducted in accordance with the manufacturers’ recommendations. It should also be noted that the software’s license is currently available.

#### 2.1.3. Additive Manufacturing of the Dental Models

The obtained digital dental models (corresponding to standard models or to real patients) can be 3D printed. The STL files were generated and exported to an SLA (Stereolithography) 3D printer (Form 3B+, Formlabs Inc., Somerville, MA, USA) in order to fabricate the printed resin models using their proprietary resin (White Resin, Formlabs Inc., Somerville, MA, USA). Printing tests were performed at a layer height of 0.050 mm, 0.100 mm and 0.020 mm; a layer height of 0.050 mm was chosen to be used in printing, combining both a reduced printing time and a proper level of detail. The resulting plates designated for printing consisted of 7 models, placed in a vertical position, to maximize the available space. Printing was carried out at a layer height of 0.050 mm, 4.5 s layer exposure time, at 60% UV power. The printing time of 7 h and 38 min allowed for multiple plates to be printed per day, significantly reducing the total production time. When the printing program had finished, the printed models were removed from the platform and first cleaned (washed) for ten minutes in 98% Isopropyl Alcohol (Anycubic Wash & Cure Plus Machine, Hongkong Anycubic Technology Co., Kowloon, Hong Kong, China) to remove any excess resin. After cleaning and drying, the printed models rested for at least 10 min to make sure that they were dry and free of ethanol residue. The printed parts were placed for 6 min for curing under UV light (Formlabs Cure Station, Formlabs Inc., Somerville, MA, USA) at ambient temperature, for final, optimal polymerization. The support structures were removed and the obtained 3D-printed dental models did not require any further finishing or polishing. In case of the modular dental model, this can be printed with or without mobile dies/abutments. It is worth noting that the dental models’ bases were designed and 3D printed so that they can be mounted (using a magnetic metal plate) in the dental patient simulators (Dental Patient Simulator (DPS) “Adam” TM, KaVo Dental GmbH, 88400 Biberach/Riß, Germany) which the faculty makes available to students.

### 2.2. Evaluation of Undergraduate Students’ Opinion on the Use of Modular Digital and 3D-Printed Dental Models in University Dental Education

#### 2.2.1. Survey Methodology and Ethical Approval

This survey was approved by the Scientific Research Ethics Committee of “Carol Davila” University of Medicine and Pharmacy, Bucharest, Romania (Project identification Code: PO-35-F-03; Protocol number:15596; date: 8 June 2022). The students’ opinion on the use of modular digital dental models and 3D-printed simulated dental models in university dental education was assessed via a questionnaire, administered as a Google form sent via email. The study was conducted in accordance with the Declaration of Helsinki of 1975, revised in 2013. Subjects selected to participate in the study were invited to fill in the questionnaire and were informed about the survey in accordance with the World Medical Association Declaration of Helsinki and the current European privacy law, highlighting, in an introduction section of the questionnaire, the scientific aim of the study, that the questionnaire was anonymous and that they had the right to interrupt the completion of the form at any moment in case of withdrawal. Subjects who were invited to participate in this study received an e-mail containing written information about the study and the informed consent. All subjects gave their informed consent for inclusion before they participated in the study. All subjects that agreed to participate in the study expressed their consent by completing the survey; informed consent was obtained from the subjects involved in the study to publish this paper. No personal data were collected through the form and, as an anonymous web survey, no sensitive data were collected. The questionnaire was secured so as to be completed only once by every participant.

#### 2.2.2. Selection of Participants

This study was designed to be a pilot study and was conducted among third- and fourth-year dental students undergoing theoretical and practical training in Prosthodontics, Faculty of Dentistry, “Carol Davila” University of Medicine and Pharmacy, Bucharest, Romania. The inclusion criteria for the participants were: third- and fourth-year dental students; students that completed the theoretical and practical training in Fixed Prosthodontics and Occlusology, for an entire semester; students that used the modular digital and 3D-printed dental models in their training activities, during an entire semester (14 weeks), at least once per week; females or males aged >18 years. The exclusion criteria were: students not willing to participate in the study; students that have not completed their training in Fixed Prosthodontics and Occlusology for an entire semester. These students’ opinions on the use of modular digital dental models and 3D-printed simulated dental models in university dental education was assessed via a questionnaire, administered as a Google form sent via email. The request to participate in the survey was applied to 240 students, of which 205 voluntarily agreed to participate, which means a percentage of acceptance of participation of 85.42%.

#### 2.2.3. Survey Questionnaire

The questionnaire used for the assessment (as presented in Table 1) was formed of 20 items represented by both multiple or single-choice questions, referring to the following main aspects: (1) socio-demographic data (age, gender, year of study—first three questions of the questionnaire); (2) participants’ perception of the proposed computerized method allowing virtual case simulation on modular digital dental models (questions 4 to 10); (3) participants’ perception of the practical training on 3D-printed dental models (questions 11 to 17); (4) participants’ perception of certain particularities of the proposed methods and on their further use in university dental education (questions 18 to 20). Items 4 to 7 and 11 to 14 were represented by multiple-choice questions; the other items were represented by single-choice questions. The estimated time needed to fill in the questionnaire was a maximum of 5 min.

#### 2.2.4. Data Analysis

All the data from the study were analyzed using IBM SPSS Statistics 25 and illustrated using Microsoft Office Excel/Word 2021. Quantitative variables were tested for normal distribution using the Shapiro–Wilk Test and were summarized as averages with standard deviations or medians with interquartile ranges. Qualitative variables were reported as counts or percentages and differences between groups were tested using Fisher’s Exact tests. Quantitative independent variables with non-parametric distributions were tested between groups using Mann–Whitney U. Correlations between quantitative variables with non-parametric distributions were measured using Spearman’s rho correlation coefficients.

## 3. Results

### 3.1. Modular Digital and 3D-Printed Dental Models

Theoretical and hands-on courses on the use of modular digital dental models and of 3D-printed simulated dental models were organized weekly by experienced specialists in prosthodontics (university teaching staff) for the third- and fourth-year dental students, and the students used these models in their training activities during an entire semester (14 weeks) at least once per week. The modular digital models were used by the dental students under the supervision of the teaching staff. The students could perform repeated virtual simulation of any possible partial edentulism scenario, the virtual identification of classes of partial edentulism requiring prosthetic treatment, the digital analysis of diverse partial edentulous cases, occlusal examination (i.e., malpositioned teeth, occlusal plane) and the selection of the proper dental preparation and of fixed dental prostheses for different virtual clinical situations.

In Figure 9, Figure 10, Figure 11, Figure 12 and Figure 13 certain sequences from the development of a modular digital dental model are presented; this specific model was classified by our teaching staff as a “training digital dental model”. The “training digital dental model” was designed to be relevant to undergraduate students in fixed prosthodontics, as it presented different edentulous areas, migrated teeth and bone resorptions. Based on this “digital training dental model”, “3D-printed training dental models” were subsequently manufactured. The previously mentioned 3D-printed models were made available to both third-year and fourth-year dental students, who performed hands-on practice on these models throughout an entire semester. Students’ practical training on these 3D-printed models included procedures related to fixed prosthodontics, such as teeth preparations, wax-ups, fabrication of interim restorations, partial and full-arch impressions, evaluation and provisional cementation of interim fixed prosthetic restorations. Additionally, 3D-printed dental models that presented diverse ideal teeth preparations for prosthetic restorations were made available to students; students could check their own dental preparations using these models as reference. During the respective semester, the students succeeded in combining virtual training on modular digital dental models with practical training on their corresponding 3D-printed models.

### 3.2. Students’ Opinion on the Use of Modular Digital and 3D-Printed Dental Models in University Dental Education

The statistical analysis of the results obtained from the applied survey revealed the following presented aspects. Data in Table 2 show the demographic characteristics of the analyzed students. Mean age was 22.31 ± 1.74, with a median of 22 years. Most of the students were women (74.6%). A 58% proportion of the students were in their fourth academic year and 42% of the students were in their third academic year.

Data in Table 3 show the distribution of the students according to their answers in the survey. The results show the following elements:–the most-selected answer for the functions of modular digital dental models (question 4) was the realistic simulation of various classes of partial edentulism (81%); corresponding to functions of 3D-printed dental models (question 11), the most-selected answer was option A (improvement of practical skills) (78.5%);–the most-selected answer for advantages of modular digital dental models in university training (question 5) was option A (3D visualization of details) (85.9%); in the same line, the most-selected answer for the advantages of 3D-printed dental models in university training (question 12) was option A (real/3D visualization of details) (82.9%);–the most-selected answer for advantages of modular digital dental models as a method of e-learning (question 6) was option A (easy access from various locations) (86.8%); in the case of 3D-printed dental models, the most-selected advantage was that the use of these 3D-printed models allows direct feedback to students from teachers (77.1%); –the most-selected answer for the disadvantages of modular digital dental models as a method of e-learning (question 7) was option D (limitation of direct interaction with patients) (69.3%); as for the 3D-printed dental models, the most-selected answer for their disadvantages was option B (hardness of the printed material is different from that of natural teeth) (91.7%);–most responses were affirmative to questions 8 (76.6%), 9 (82.9%) and 10 (91.7%),which are related to the virtual simulation method, and to questions 15 (84.9%), 16 (90.2%), and 17 (92.7%), which are related to the 3D-printed dental models. The students were asked if they feel better-prepared for their clinical activity by using the proposed methods (question 8 and 15), if the methods fits their way of learning (question 9 and 16) and if they are interested in further use of the proposed methods in university training (question 10 and 17). It can be noted that the questions related to 3D-printed dental models registered a slightly higher affirmative response percentage than the questions related to the virtual simulation method (modular digital models);–most of the students responded affirmatively to question 18 (obtaining virtual/3D-printed models from real clinical cases is an advantage) (96.6%), question 19 (usage of virtual/3D-printed models improves professional skills in digital technology and 3D-printing) (92.7%), and question 20 (development of virtual/3D-printed dental models would be of interest to future generations of students) (96.6%).

Data from Table 4 show the comparison of ages between different question item answers. The distribution of age between groups was non-parametric according to the Shapiro–Wilk test (*p* < 0.001). According to the Mann–Whitney U test, students who selected item 5D (modular digital dental models are a quick way to learn) had a significantly lower age (median = 22 years, IQR = 21–23) than students who did not select 5D (median = 22 years, IQR = 22–23) (*p* = 0.034).

Also, students who selected item 5E (modular digital dental models are a comfortable way of learning) (median = 22 years, IQR = 22–23 vs. median = 22 years, IQR = 21–23, *p* = 0.048), 6E (modular digital dental models allow the evaluation of the program’s effectiveness) (median = 23 years, IQR = 22–23 vs. median = 22 years, IQR = 21–23, *p* = 0.008) or 13B (3D-printed dental models allow direct interaction between students and teachers) (median = 22 years, IQR = 22–23 vs. median = 22 years, IQR = 21–23, *p* = 0.022) had a significantly higher age than students who did not select these items.

Data from Table 5 show the distribution of the students according to academic year of study and item answers. The results show the following:–Students who selected option 4A (modular digital dental models allow realistic simulation of various classes of partial edentulism) were more frequently in their fourth academic year (85.7% vs. 74.4%) (*p* = 0.048);–Students who selected option 4C (modular digital dental models allow realistic simulation of alveolar ridge resorption) were more frequently in their fourth academic year (65.5% vs. 51.2%) (*p* = 0.044);–Students who selected option 4D (modular digital dental models allow realistic simulation of malpositioned teeth and destruction of dental crowns) were more frequently in their fourth academic year (76.5% vs. 61.6%) (*p* = 0.030);–Students who selected option 5D (modular digital dental models are a quick way to learn) were more frequently in their third academic year (46.5% vs. 27.7%) (*p* = 0.008);–Students who selected option 7C (modular digital dental models are limited in terms of direct interaction with teachers) were more frequently in their fourth academic year (64.7% vs. 47.7%) (*p* = 0.022);–Students who answered that modular digital dental models fit their way of learning (question 9) were more frequently in their third academic year (89.5% vs. 78.2%) (*p* = 0.039);–Students who selected option 12D (3D-printed dental models are a quick way to learn) were more frequently in their third academic year (45.3% vs. 28.6%) (*p* = 0.018);–Students who selected option 13B (3D-printed dental models allow direct interaction between students and teachers) were more frequently in their fourth academic year (59.7% vs. 40.7%) (*p* = 0.011);–Students who answered that virtual/3D-printed dental models would be of interest to future generations of students (question 20) were more frequently in their third academic year (100% vs. 94.1%) (*p* = 0.043).

Data from Table 6 show the comparisons of item answers among the analyzed students. Differences between groups that are significant according to Fisher’s Exact Tests (*p* < 0.05) show the following:–Students who selected item 5A (modular digital dental models allow 3D visualization of details) more frequently also selected item 12A (3D-printed dental models allow 3D visualization of details) (90.6% vs. 62.9%) (*p* < 0.001);–Students who selected item 5B (modular digital dental models allow repeated virtual simulations) more frequently also selected item 12B (3D-printed dental models allow repeated attempts of various practical procedures) (81.9% vs. 36.1%) (*p* < 0.001);–Students who selected item 5C (modular digital dental models are an accessible and flexible method of learning) more rarely also selected item 12C (3D-printed dental models are an accessible and flexible method of learning) (78.8% vs. 43.8%) (*p* < 0.001);–Students who selected item 5D (modular digital dental models are a quick way to learn) more frequently also selected item 12D (3D-printed dental models are a quick way to learn) (68.5% vs. 17.4%) (*p* < 0.001);–Students who selected item 5E (modular digital dental models are a comfortable way of learning) more frequently also selected item 12E (3D-printed dental models are a comfortable way of learning) (67% vs. 26.5%) (*p* < 0.001);–Students who selected item 6C (modular digital dental models allow fast virtual feedback to students from teachers) more frequently also selected item 13C (3D-printed dental models allow direct feedback to students from teachers) (60.8% vs. 42.6%) (*p* = 0.030);–Students who answered that modular digital models are a good way to prepare for their clinical activity (question 8) more frequently also answered that 3D-printed dental models are a good way to prepare for their clinical activity (question 15) (83.3% vs. 38.7%) (*p* < 0.001);–Students who answered that modular digital models fit their way of learning (question 9) more frequently also answered that 3D-printed dental models fit their way of learning (question 16) (87.6% vs. 40%) (*p* < 0.001);–Students who answered that they are interested in further use of modular digital models (question 10) more frequently also answered that they are interested in further use of 3D-printed dental models (question 17) (95.3% vs. 46.7%) (*p* < 0.001);–Students who answered that obtaining virtual/3D-printed dental models from real clinical cases is an advantage (question 18) more frequently also answered that virtual/3D-printed dental models can help them improve their professional skills in digital technology and 3D-printing (question 19) (98.9% vs. 66.7%) (*p* < 0.001);–Students who answered that obtaining virtual/3D-printed dental models from real clinical cases is an advantage (question 18) more frequently also answered that development of virtual/3D-printed dental models would be of interest to future generations of students (question 20) (98.5% vs. 42.9%) (*p* < 0.001);–Students who answered that virtual/3D-printed dental models can help them improve their professional skills in digital technology and 3D-printing (question 19) more frequently also answered that development of virtual/3D-printed dental models would be of interest to future generations of students (question 20) (94.9% vs. 28.6%) (*p* < 0.001).

Based on the numbers of items chosen for question 4 a score was constructed: Score_Q4—number of functions chosen for modular digital dental models. Data from Table 7 and Figure 14 show the comparison of Score_Q4 between academic years of study. Based on the numbers of items chosen for questions 4 and 11 scores were constructed: Score_Q4—number of functions chosen for modular digital dental models, Score_Q11—number of functions chosen for 3D-printed dental models. Distribution of the Score_Q4 between groups was non-parametric according to the Shapiro–Wilk test (*p* < 0.001). According to the Mann–Whitney U test, fourth year students selected significantly more functions for modular digital dental models (median = 4, IQR = 3–5) than third year students (median = 3, IQR = 2–5) (*p* = 0.046).

Data from Table 8 show the correlation between Score_Q4 and Score_Q11. Both scores have a non-parametric distribution according to the Shapiro–Wilk Test (*p* < 0.001). The correlation observed is statistically significant and positive with a weak power (*p* < 0.001, R = 0.242), indicating that students who chose more functions for modular digital dental models also chose more functions for 3D-printed dental models.

## 4. Discussion

Digital technology is already advanced in the medical domain, and has an extraordinary pace of and potential for development [36,37]. As regards the educational area in dentistry, new digital means for theoretical teaching and practical training have appeared in recent years, as well as various modalities for virtually interacting with students or testing their abilities.

Based on the digitization of standard dental models, of artificial intact teeth, and of artificial prepared teeth for diverse types of prosthetic restorations, the virtual method presented in this paper allows the following: simulation of any possible partial edentulism scenario; digital identification of diverse classes of partial edentulism; occlusal examination; establishment of appropriate prosthetic treatment plans for different virtual clinical situations; and selection of the proper dental preparation and fixed dental prosthesis. In light of the published literature, as other authors have also shown, this type of digital training improves students’ 3D visualization skills while enabling them to better understand and assimilate the required didactic information [22,38,39,40]. The virtual partial edentulism cases that can be designed by this method correspond to clinical situations that a dentist encounters in daily practice. Furthermore, based on the designed modular digital model, dental educators can add details inspired from real clinical cases to their interactive discussions with their students, including anatomical variations or ectopic dental positions. Nevertheless, this method also allows the creation of modular digital models based on real clinical cases. The digital library created by the digitalization of standard dental models and of real clinical cases allows virtual simulations of any possible partial edentulism scenario; in our opinion, this enables us to enrich our faculty’s own digital library, which could be of great benefit to our students. The proposed method also allows the 3D printing of dental simulated models, which have already proved their usefulness in undergraduate students’ pre-clinical practical training, as other scientific studies have pointed out [18,30,31,32,33,41,42,43,44]. Certain authors have suggested that 3D-printed dental models represent a “more realistic and cost-efficient alternative to commercial models” in undergraduate dental training [42]. Our presented method is, indeed, software-based and available to all licensed users; however, to our understanding, this method could be regarded as being accessible, convenient and user friendly. Moreover, in our study, the 3D-printed simulated dental models were manufactured in our university, using our own internal resources, which leads to reduced fabrication costs.

Regarding the use of the proposed modular digital dental models, the possibility to constantly and rapidly vary the clinical scenarios but also to return to certain elements of interest at any time could result in the enhancement of students’ ability to promptly identify classes of partial edentulism, establish diagnoses and fixed or removable prosthetic treatment plans, and observe the succession of therapeutic stages, as other authors also stated in their studies [22,45]. As a result, students may be stimulated to elaborate several 3D virtual treatment plans for the same clinical situation, previsualize (repeatedly, if needed) these therapeutic alternatives and acknowledge the pros and cons of each possible solution. Other recent studies [46,47], whose observations support the method we put forward in our study, confirm students’ appreciation for e-learning or virtual teaching; the authors stated that e-learning offers easy access to the shared materials and feasibility, which help students better understand complex clinical cases or theoretical subjects. Mladenovic et al. [48] also reported that students appreciate the flexibility of a suggested mobile application for the dynamic study of dental traumatology. The digital method proposed in this paper also enables the computerized analysis of the discrete differences between various types of dental preparations: the unique details of each preparation can be 3D- and computer-visualized. Goodacre [40] reports that, thanks to the 3D education in dentistry, students manage to enhance their ability to visualize diverse structures three dimensionally and operate them in their minds (“spatial ability”). Other studies state, in the light of the results obtained, the necessity of virtual instruments for controlling the dental preparations achieved by the students [22,49,50]. As other authors have mentioned, students can assimilate, by means of this digital training, the notions that are important for an adequate presentation of the treatment plan to future patients, and thus improve their communication and relational skills, aspects that other authors refer to as well [21,51].

On the other hand, the idea of combining 3D virtual models with 3D-printed dental models in university dental education is also suggested by other authors [44] in order to ease visual recognition and understanding of teeth preparation. In our study, the possibility to 3D-print the dental models on the basis of the designed digital modular models obtained via the suggested method opens up new directions for applicability in dental education. The obtained 3D-printed simulation models are hence designated for hands-on training (tooth preparation for fixed prosthetic restorations or fillings, impressions, wax-up procedures, provisional restorations, evaluation and cementation of fixed prosthetic restorations) or for exercising other specific practical techniques. We have noticed that the possibility to constantly and rapidly return to certain elements of interest in the digital models allowed our students to manage the procedures they performed on the 3D-printed models. These 3D-printed dental models are also useful for additional analysis of the elements specific to edentulism cases or for establishing treatment plans and the succession of the treatment stages in dental prosthetics, occlusology and in other specialties (implantology, maxillofacial prosthodontics, orthodontics) [11,52,53,54,55]. Scientific studies found in the specialized literature reveal the contribution that teaching methods applying 3D-printed models have in dental education, as follows. Lambrecht (2010) [56] achieved 3D-printed training models for implantology surgery, which were fabricated on the basis of real patients’ CBCT. Soares (2013) [44] created virtual models and prototypes of teeth with various cavity preparations. Kroger (2017) [57] obtained 3D-printed models by digitally scanning real patients. Hohne (2019) [58,59] created models in which the enamel and dentine can be discerned as well as 3D-printed models with caries (both studies were evaluated and validated using questionnaires filled in by students). Boonsiriphant (2019) [60] printed models with ideal preparations of teeth and reported high advantages in students’ achieving practical abilities thanks to real 3D visualization in contrast to 2D images in books or courses. Werz (2018) [61] and Hanish (2020) [62] obtained printed models for exercising surgery techniques (sinus lift, extraction of wisdom teeth, apical resection). Lee (2015) [63] demonstrated that printing technology can also be useful in orthodontics, while Marty (2018) [64] presented a comparison between students’ perception on 3D-printed models based on real cases and their perception on standard models, in pedodontics.

Furthermore, the responses collected from the survey of third- and fourth-year dental students in our faculty indicate that the proposed method (the combined use of modular digital and 3D-printed dental models in university dental training) received good feedback from the participants in the survey. The findings of our survey were consistent with those of other studies found in the dental literature [65,66,67,68,69,70]. However, to our knowledge, few scientific studies have been dedicated to topics related to the influence of students’ age or academic year on their perception of the use of digital dentistry and technological advancement such as 3D-printing in dental education. Certain aspects were revealed through the statistical analysis that was performed in our study; for example, the opinion of the students participating in the study is different depending on their age: students who selected the item indicating that “modular digital dental models are considered a quick way to learn” had a significantly lower age than students who did not select this item (*p* = 0.034). When compared to younger students, older students were more interested in academic, teaching aspects, valuing the fact that modular digital dental models allow the evaluation of the method’s effectiveness (*p* = 0.008) (number of registered downloads) and that 3D-printed dental models allow direct interaction between students and teachers (*p* = 0.022). Significant differences were registered between the students’ responses to the applied questionnaire, depending on their academic year of study. Thus, when comparing the fourth-year students to the third-year students, the latter responded more frequently that modular digital dental models are a quick way to learn (*p* = 0.008) and fit their way of learning (*p* = 0.039), and that the 3D-printed dental models allow them to learn quicker (*p* = 0.018). On the other hand, fourth-year students showed greater concerns towards exploring detailed anatomical elements: fourth-year students selected more frequently the items indicating that modular digital dental models allow realistic simulation of various classes of partial edentulism (*p* = 0.048), of alveolar ridge resorption (*p* = 0.044) and of malpositioned teeth or destruction of dental crowns (*p* = 0.030). Moreover, fourth-year students selected significantly more functions for modular digital dental models than third-year students; students who chose more functions for modular digital dental models also chose more functions for 3D-printed dental models (*p* < 0.001). To our understanding, the older fourth-year students are able to better discern the value of the proposed modular digital and 3D-printed dental models because they have greater practical experience and higher theoretical knowledge. Even though the fourth-year students partake in various clinical training sessions, they have expressed their willingness to continue using the modular digital and 3D-printed dental models in their university education. 

Although we registered certain relevant differences in students’ perception of the use of the advanced method, depending on their age and academic year of study, it is worth noting that the performed statistical analysis confirms that there is a coherent pattern in students’ answers. For example, the questionnaire’s items corresponding to the modular digital models were answered similarly to those corresponding to the 3D-printed models; students gave balanced answers to the questions regarding the advantages and the usage of these two types of dental models. We can consequently conclude that the students participating in this study appreciated the value of both modular digital and 3D-printed dental models in a similar manner.

The survey represents an important tool which could contribute to the improvement of students’ training activity, especially when modern, recent innovations such as CAD/CAM technology or digital dentistry are introduced in a curriculum. Moreover, we consider that our study lines up with the idea of sustainability in university dental education which has been at the center of recent scientific studies [70,71,72,73]. In addition, other studies pointed out that, in the context of the recent COVID-19 pandemic evolution [74,75,76,77,78], hybrid teaching and educational platforms have proved to be very useful; these issues back up our proposed method.

As other authors have already pointed out [65,79] it is debatable whether digital learning methods improve dental students’ skills. Despite the noticeable, evident enthusiasm of the undergraduate dental students toward digital technology that was highlighted in this paper, structured self-learning and self-evaluation of students represent important issues for further development of dental curricula, as other authors have indicated [68]. On the other hand, implementation of dental technology in university dental education could face diverse barriers such as limited resources, funds, time and availability of teaching staff [67]. Therefore, as regards the limits of the advanced digital method, it is worth noting that it requires technological resources and certain digital skills for the students. Additionally, the hardness and the color of the printed resin that we used to obtain the 3D-printed dental models is different from that of natural teeth, as the students also noticed; in this context, the properties of different resin materials used in manufacturing the 3D-printed dental models will be a subject of our further research. Another limitation of our paper is the fact that the questionnaire was conducted at a single dental school and in a single department (Prosthodontics); therefore, its results may not be extended to students in other programs or to other dental specialties. Moreover, the results of the applied questionnaires are based on perceived experiences and not on objective evaluations; we thus consider that the development of students’ competence, skills and self-assessment should be investigated from more of an objective standpoint in future studies.

Nevertheless, our study, along with other recent ones [18,40,74,80,81], further stimulates interest in developing scientific research in the area of digital dentistry education and also in investing in technological innovations [37,82,83,84], in order to provide the proper, sustainable education to the next generations of dentists.

## 5. Conclusions

The present paper advances an alternative digital proposal dedicated to dental education of students in the domain of prosthodontics, allowing the creation digital modular dental models corresponding to various clinical situations of partial edentulism and to subsequently obtain 3D-printed dental models that can be used for students’ practical training. The suggested method stimulates students to project, create, previsualize and interact with modular didactic digital models and to perform repeated virtual simulation of any possible partial edentulism scenario; on the other hand, the 3D-printed models offer the possibility to enhance students’ practical skills.As we registered positive feedback from students participating in the survey, the proposed method could offer students at the pre-clinical stage of their education the opportunity to train and prepare themselves better for their future clinical activities.The proposed method could pave the way for various practical training applications in dental education, fostering its sustainability and encouraging interdisciplinary collaboration.

## Figures and Tables

**Figure 1 medicina-59-00116-f001:**
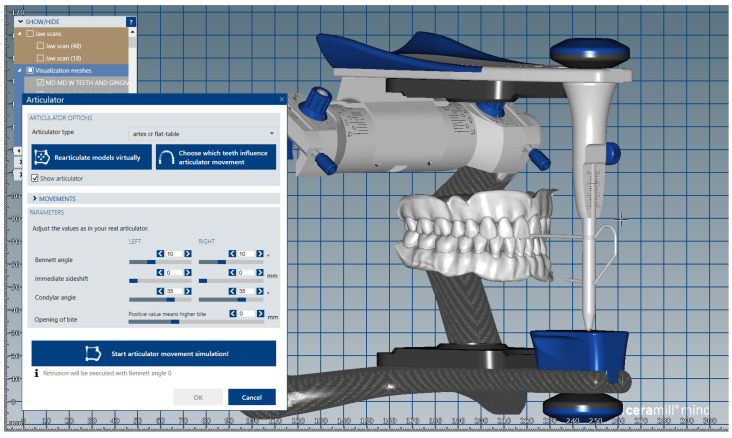
Image of digital dental model mounted in the virtual articulator in maximum intercuspation position.

**Figure 2 medicina-59-00116-f002:**
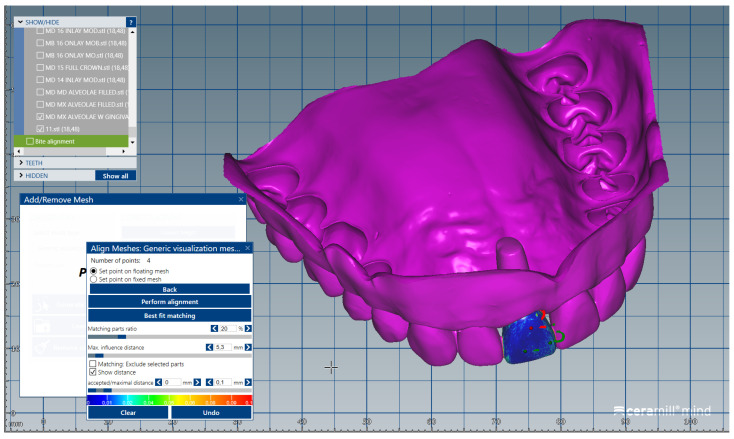
Image representing the .stl file obtained after separate scanning of the unprepared tooth # 1.1 (crown and radicular extension) indexed by reference to the corresponding surfaces of the digital dental model.

**Figure 3 medicina-59-00116-f003:**
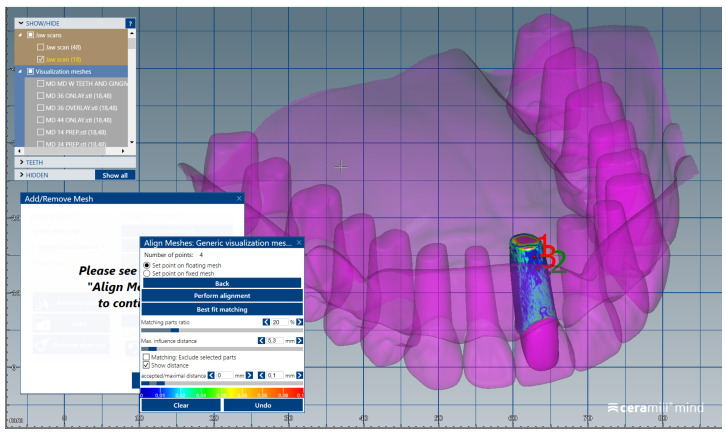
Image representing the indexation of the prepared tooth #1.1, by reference to the corresponding surfaces of the radicular extension.

**Figure 4 medicina-59-00116-f004:**
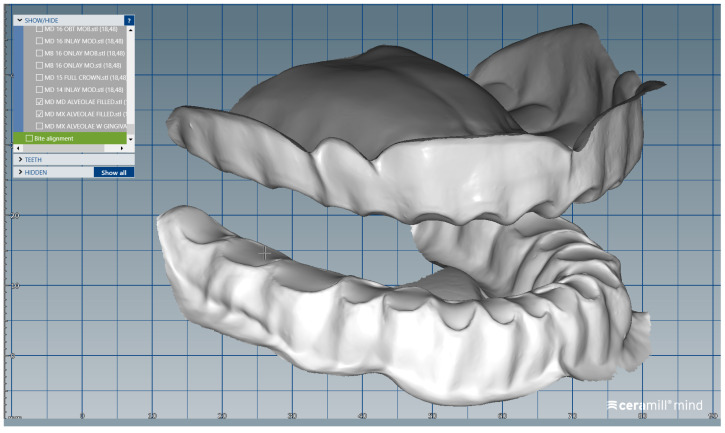
Image of the upper and lower dental alveolar processes obtained using the ”Edit Mesh” tool.

**Figure 5 medicina-59-00116-f005:**
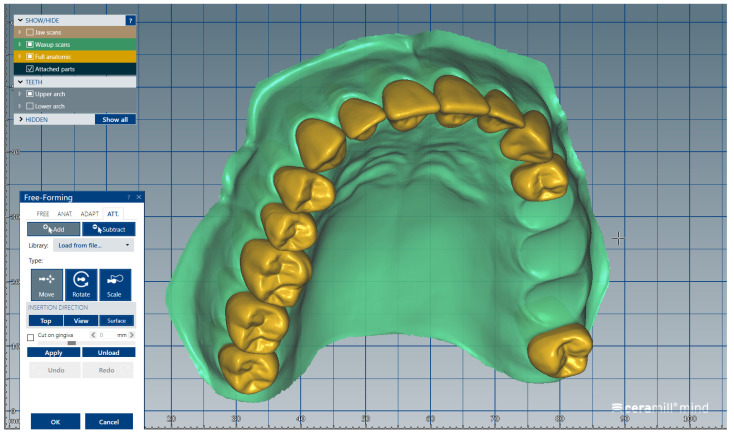
Image representing the .stl files of the unprepared teeth from #1.8 to #2.4 and tooth #2.8 that are added to the .stl file of the maxillary alveolar process (“MX GINGIVA FILLED.stl” file) as Boolean additions.

**Figure 6 medicina-59-00116-f006:**
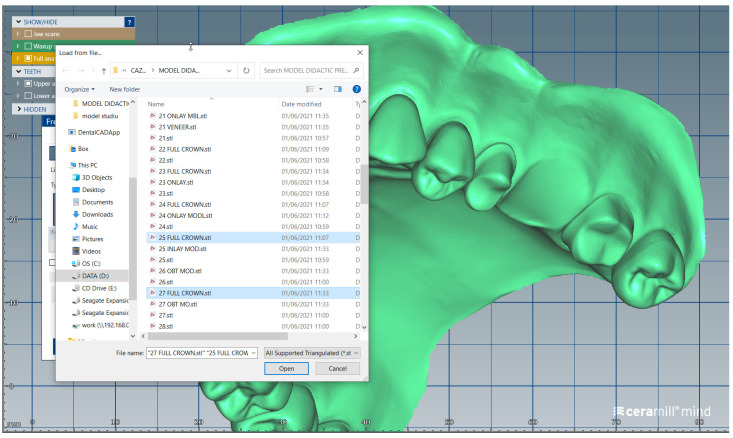
Image representing the .stl files of the prepared teeth #25 and #27 that are added to the .stl file of the maxillary alveolar process (“MX GINGIVA FILLED.stl” file) as Boolean additions.

**Figure 7 medicina-59-00116-f007:**
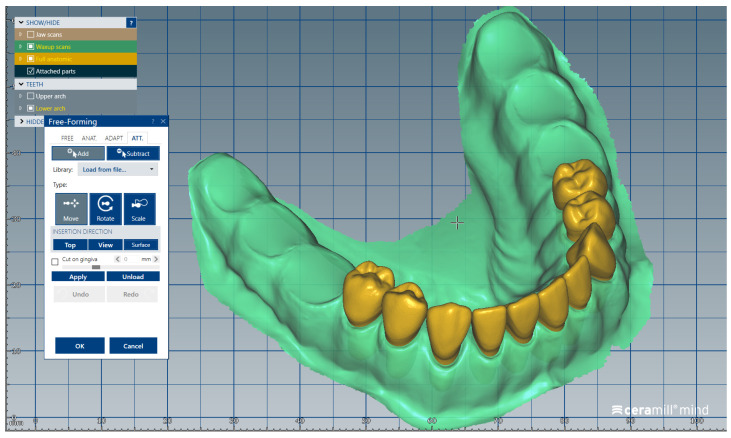
Image representing the .stl files of the unprepared teeth from #3.5 to #4.5 that are added to the .stl file of the mandibular alveolar process (“MD GINGIVA FILLED.stl” file) as Boolean additions.

**Figure 8 medicina-59-00116-f008:**
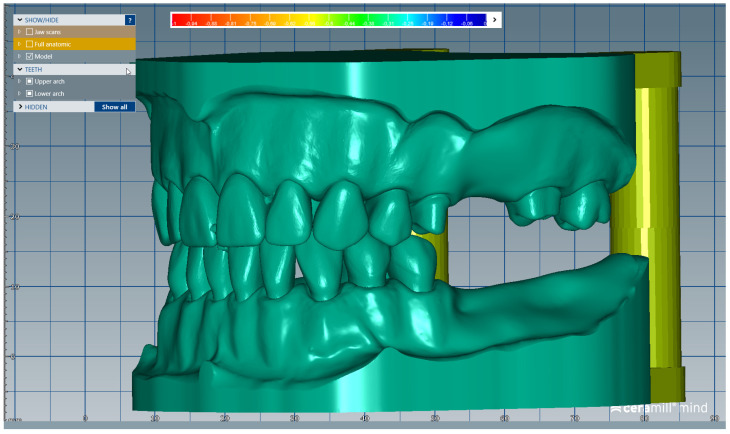
The obtained modular didactic dental model; the areas corresponding to edentulous ridges can be edited to simulate bone resorption.

**Figure 9 medicina-59-00116-f009:**
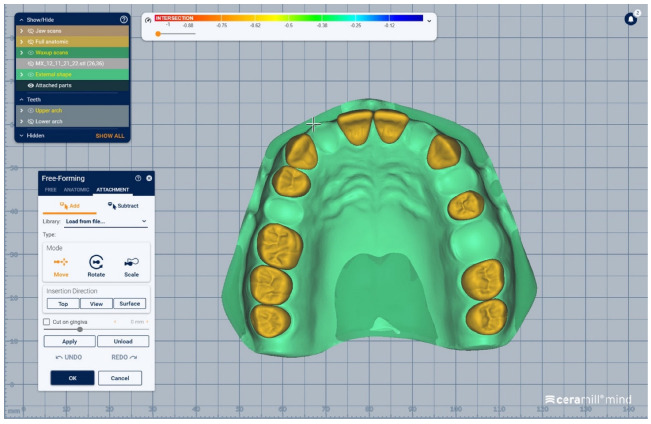
Sequence for obtaining the “training digital dental model”: the .stl files of the unprepared teeth are added to the .stl file of the maxillary alveolar process (“MX GINGIVA FILLED.stl” file) as Boolean additions.

**Figure 10 medicina-59-00116-f010:**
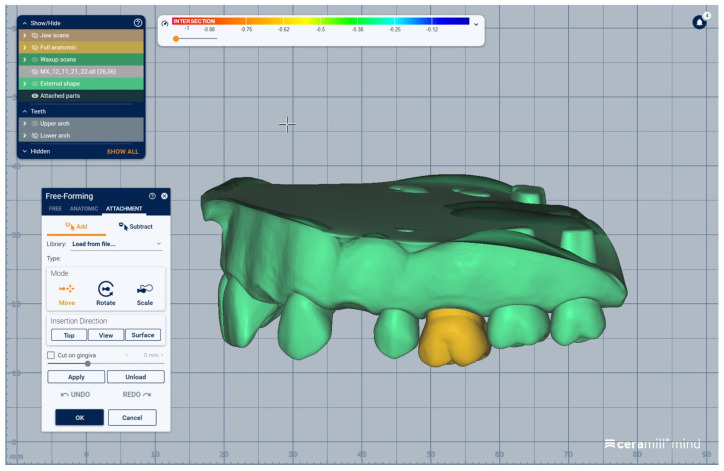
Sequence for obtaining the “training digital dental model”: the migrated tooth (#2.6) is separately added.

**Figure 11 medicina-59-00116-f011:**
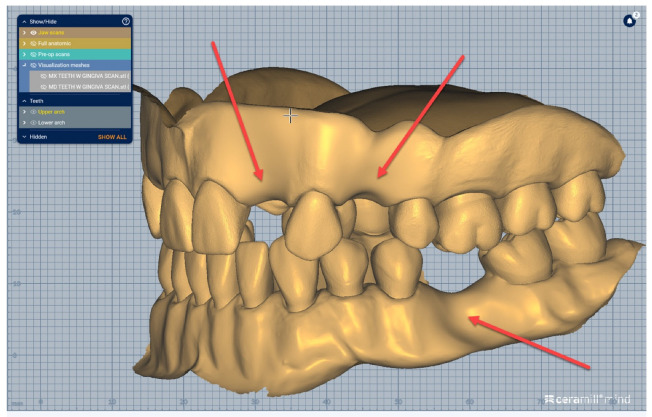
Sequence for obtaining the “training digital dental model”: the areas corresponding to edentulous ridges are edited to simulate bone resorption (indicated by red arrows).

**Figure 12 medicina-59-00116-f012:**
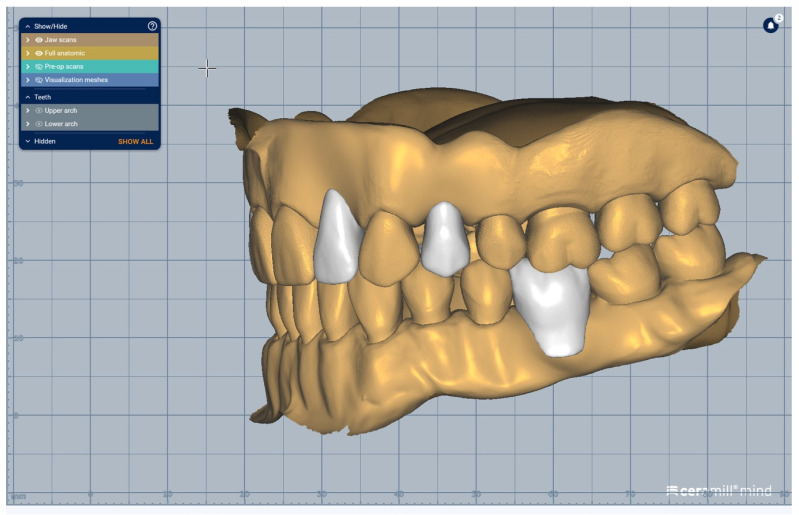
Sequence for obtaining the “training digital dental model”: As an additional option, the absent teeth can be added in order to observe certain aesthetic and functional challenges.

**Figure 13 medicina-59-00116-f013:**
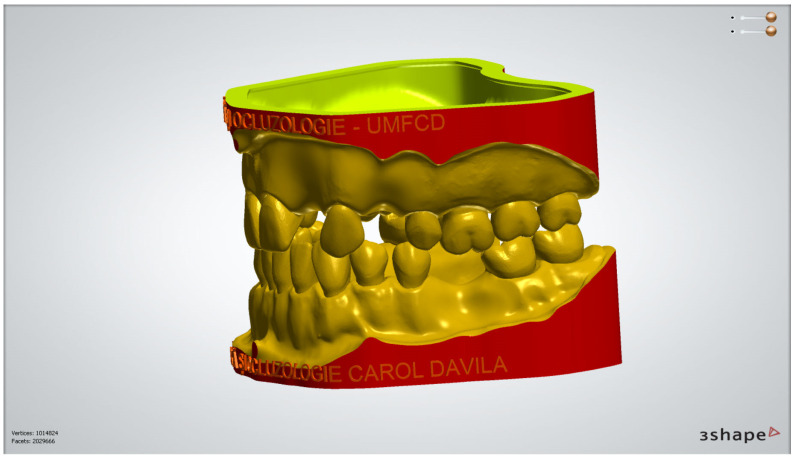
The corresponding printable digital model is obtained.

**Figure 14 medicina-59-00116-f014:**
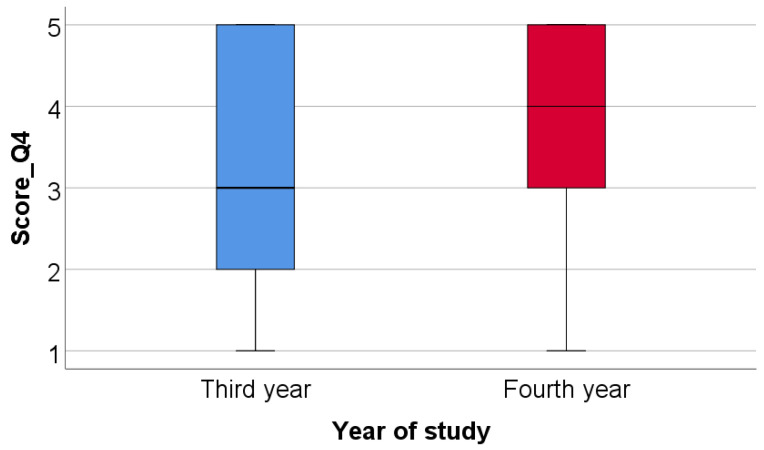
Comparison of Score_Q4 between academic years of study.

**Table 1 medicina-59-00116-t001:** The questionnaire used to assess the opinions of dental students on the use of modular digital and 3D-printed dental models in university dental education; the questionnaire was administered as a Google form and sent via email.

Investigated Aspects	Questions (Q) and Possible Answers
**(1) Socio-demo-graphic data**	**Q1.** Please enter your age
	**Q2.** Please enter your gender
	**Q3.** Please enter your study year
**(2) Modular digital dental models**	**Q4.** Modular digital dental models allow the following:
	a. realistic simulation of various classes of partial edentulism
	b. easy identification of various classes of partial edentulism
	c. realistic simulation of the alveolar ridges resorption of various degrees
	d. realistic simulation of malpositioned teeth and the destruction of dental crowns, of various degrees
	e. easy elaboration of various treatment plans
	**Q5.** The three main advantages of using modular digital dental models in my university training are as follows:
	a. it allows a 3D visualization of details, in contrast to 2D images
	b. it allows repeated virtual simulations (reiteration of virtual simulations)
	c. it is an accessible and flexible method of learning
	d. it is a quick way to learn
	e. it is a comfortable way of learning
	**Q6.** The three main advantages of using modular digital models as a method of e-learning in university dental education are as follows:
	a. it allows easy access from various locations
	b. it allows virtual interaction (synchronous and asynchronous) between students and teachers
	c. it allows fast virtual feedback to students from teachers
	d. it allows the storage of digital data for a long time
	e. it allows the evaluation of the program’s effectiveness (number of registered downloads)
	**Q7.** The three main disadvantages of using modular digital models as a method of e-learning in university dental education are as follows:
	a. it requires technological resources (dedicated electronic devices: computer, laptop etc.)
	b. it requires (minimum) experience in the field of computers/(minimum) digital skills
	c. limitation of direct interaction with teachers (face-to-face contact)
	d. limitation of direct interaction with patients
	e. dependence on internet connection
	**Q8.** Using the proposed virtual simulation method makes me feel better prepared for my clinical activity:
	a. Yes
	b. No
	**Q9.** The proposed virtual simulation method fits my way of learning:
	a. Yes
	b. No.
	**Q10.** I am interested in further use of the proposed virtual simulation method in my university training:
	a. Yes
	b. No
**(3) 3D-printed dental models**	**Q11.** Practical training on 3D-printed dental models allows me the following:
	a. to improve my practical skills
	b. to learn diverse practical procedures risk-free
	c. hands-on training under the supervision of teachers
	d. to better understand the performed procedures (dental preparations, impressions, wax-up procedures, interim restorations)
	e. good visualization of teeth (position, destruction) and edentulous areas
	**Q12.** The three main advantages of using 3D-printed dental models in my university training are as follows:
	a. it allows a real/3D visualization of the details, in contrast to the 2D images
	b. it allows repeated attempts of various practical procedures
	c. it is an accessible, flexible method of learning
	d. it is a quick way to learn
	e. it is a comfortable way of learning
	**Q13.** The three main advantages of using 3D-printed dental models in university dental education are as follows:
	a. 3D-printing of models is facilitated by the university
	b. it allows direct interaction between students and teachers
	c. it allows direct feedback to students from teachers
	d. 3D-printed models can be scanned and archived as digital models which allow virtual evaluation
	e. 3D-printed models can be used for practical training in various dental specialties in different years of study
	**Q14.** The three main disadvantages of using 3D-printed dental models in my university dental training are as follows:
	a. the colour of the printed material is different from that of natural teeth
	b. the hardness of the printed material is different from that of natural teeth
	c. the lightness of the printed material is different from that of natural teeth
	d. the absence of a gingival mask
	e. the 3D-printed models are brittle
	**Q15.** Practicing on 3D-printed dental models makes me feel better prepared for my clinical activity:
	a. Yes
	b. No
	**Q16.** The practical training on 3D-printed dental models fits my way of learning:
	a. Yes
	b. No
	**Q17.** I am interested in further use of 3D-printed dental models in my university training:
	a. Yes
	b. No
**(4) Aspects common to both modular digital and 3D-printed dental models**	**Q18.** I believe that obtaining virtual and 3D-printed dental models from real clinical cases through the proposed methods is an advantage:
	a. Yes
	b. No
	**Q19.** I believe that the use of the proposed methods in my university training can help me to improve my own professional skills in digital technology and 3D-printing:
	a. Yes
	b. No
	**Q20.** I think that the development of these teaching / learning methods (virtual and 3D-printed dental models) would be of interest to future generations of students:
	a. Yes
	b. No

**Table 2 medicina-59-00116-t002:** Demographic characteristics of the analyzed students.

Parameter	Value
Age (Mean ± SD, Median (IQR)	22.31 ± 1.74, 22 (21–23)
Gender (No., %)	153 (74.6%) Female, 52 (25.4%) Male
Year of study (No., %)	86 (42%) Third year, 119 (58%) Fourth year

**Table 3 medicina-59-00116-t003:** Distribution of the students according to the answers from the survey.

Question	Selected/Affirmative Answer (No., %)
Q4	4A-81%, 4B-79.5%, 4C-59.5%, 4D-70.2%, 4E-78%
Q5	5A-85.9%, 5B-68.3%, 5C-66.3%, 5D-35.6%, 5E-43.9%
Q6	6A-86.8%, 6B-76.6%, 6C-56.6%, 6D-64.4%, 6E-15.6%
Q7	7A-62.4%, 7B-55.6%, 7C-57.6%, 7D-69.3%, 7E-55.1%
Q8	48 (23.4%) Negative, 157 (76.6%) Affirmative
Q9	35 (17.1%) Negative, 170 (82.9%) Affirmative
Q10	17 (8.3%) Negative, 188 (91.7%) Affirmative
Q11	11A-78.5%, 11B-75.6%, 11C-54.6%, 11D-78%, 11E-62.9%
Q12	12A-82.9%, 12B-70.2%, 12C-35.6%, 12D-35.6%, 12E-42.9%
Q13	13A-53.2%, 13B-51.7%, 13C-77.1%, 13D-52.7%, 13E-64.9%
Q14	14A-61.5%, 14B-91.7%, 14C-58%, 14D-42.4%, 14E-44.4%
Q15	31 (15.1%) Negative, 174 (84.9%) Affirmative
Q16	20 (9.8%) Negative, 185 (90.2%) Affirmative
Q17	15 (7.3%) Negative, 190 (92.7%) Affirmative
Q18	7 (3.4%) Negative, 198 (96.6%) Affirmative
Q19	15 (7.3%) Negative, 190 (92.7%) Affirmative
Q20	7 (3.4%) Negative, 198 (96.6%) Affirmative

**Table 4 medicina-59-00116-t004:** Comparison of ages between different question item answers.

Age/Item		Q5-D	Q5-E	Q6-E	Q13-B
	Average ± SD	22.48 ± 1.98	22.18 ± 1.78	22.22 ± 1.75	22.01 ± 1.07
	Median (IQR)	22 (22–23)	22 (21–23)	22 (21–23)	22 (21–23)
	Mean Rank	109.22	96.12	98.49	93.64
Selected	Average ± SD	21.99 ± 1.15	22.47 ± 1.69	22.78 ± 1.66	22.58 ± 2.16
	Median (IQR)	22 (21–23)	22 (22–23)	23 (22–23)	22 (22–23)
	Mean Rank	91.75	111.79	127.41	111.75
*p* *		0.034	0.048	0.008	0.022

* Mann–Whitney U Test.

**Table 5 medicina-59-00116-t005:** Distribution of the students according to academic year of study and item answers.

Selected Item /Year of Study	Third Year (N = 86)		Fourth Year (N = 119)		*p* *
	No.	%	No.	%	
Q4-A	64	74.4%	102	85.7%	0.048
Q4-C	44	51.2%	78	65.5%	0.044
Q4-D	53	61.6%	91	76.5%	0.030
Q5-D	40	46.5%	33	27.7%	0.008
Q7-C	41	47.7%	77	64.7%	0.022
Q9 (Affirmative)	77	89.5%	93	78.2%	0.039
Q12-D	39	45.3%	34	28.6%	0.018
Q13-B	35	40.7%	71	59.7%	0.011
Q20 (Affirmative)	86	100%	112	94.1%	0.043

* Fisher’s Exact Test.

**Table 6 medicina-59-00116-t006:** Comparisons of item answers among analyzed students.

**Item Q5-A** **/Q12-A**	**12-A-Not Selected**		**12-A-Selected**		* **p** * *****
	**No.**	**%**	**No.**	**%**	
5-A-Not selected	13	37.1%	16	9.4%	<0.001
5-A-Selected	22	62.9%	154	90.6%	
**Item Q5-B** **/Q12-B**	**12-B-Not selected**		**12-B-Selected**		* **p** * *****
	**No.**	**%**	**No.**	**%**	
5-B-Not selected	39	63.9%	26	18.1%	<0.001
5-B-Selected	22	36.1%	118	81.9%	
**Item Q5-C** **/Q12-C**	**12-C-Not Selected**		**12-C-Selected**		* **p** * *****
	**No.**	**%**	**No.**	**%**	
5-C-Not selected	28	21.2%	41	56.2%	<0.001
5-C-Selected	104	78.8%	32	43.8%	
**Item Q5-D** **/Q12-D**	**12-D-Not Selected**		**12-D-Selected**		* **p** * *****
	**No.**	**%**	**No.**	**%**	
5-D-Not selected	109	82.6%	23	31.5%	<0.001
5-D-Selected	23	17.4%	50	68.5%	
**Item Q5-E** **/Q12-E**	**12-E-Not Selected**		**12-E-Selected**		* **p** * *****
	**No.**	**%**	**No.**	**%**	
5-E-Not selected	86	73.5%	29	33%	<0.001
5-E-Selected	31	26.5%	59	67%	
**Item Q6-C** **/Q13-C**	**13-C-Not Selected**		**13-C-Selected**		* **p** * *****
	**No.**	**%**	**No.**	**%**	
6-C-Not selected	27	57.4%	62	39.2%	0.030
6-C-Selected	20	42.6%	96	60.8%	
**Item Q8** **/Q15**	**15-Negative**		**15-Affirmative**		* **p** * *****
	**No.**	**%**	**No.**	**%**	
8-Negative	19	61.3%	29	16.7%	<0.001
8-Affirmative	12	38.7%	145	83.3%	
**Item Q9** **/Q16**	**16-Negative**		**16-Affirmative**		* **p** * *****
	**No.**	**%**	**No.**	**%**	
9-Negative	12	60%	23	12.4%	<0.001
9-Affirmative	8	40%	162	87.6%	
**Item Q10** **/Q17**	**17-Negative**		**17-Affirmative**		* **p** * *****
	**No.**	**%**	**No.**	**%**	
10-Negative	8	53.3%	9	4.7%	<0.001
10-Affirmative	7	46.7%	181	95.3%	
**Item Q18** **/Q19**	**19-Negative**		**19-Affirmative**		* **p** * *****
	**No.**	**%**	**No.**	**%**	
18-Negative	5	33.3%	2	1.1%	<0.001
18-Affirmative	10	66.7%	188	98.9%	
**Item Q18** **/Q20**	**20-Negative**		**20-Affirmative**		* **p** * *****
	**No.**	**%**	**No.**	**%**	
18-Negative	4	57.1%	3	1.5%	<0.001
18-Affirmative	3	42.9%	195	98.5%	
**Item Q19** **/Q20**	**20-Negative**		**20-Affirmative**		* **p** * *****
	**No.**	**%**	**No.**	**%**	
19-Negative	5	71.4%	10	5.1%	<0.001
19-Affirmative	2	28.6%	188	94.9%	

* Fisher’s Exact Test.

**Table 7 medicina-59-00116-t007:** Comparison of Score_Q4 between academic years of study.

Year of Study/Score_Q4	Average ± SD	Median (IQR)	Mean Rank	*p* *
Third year	3.42 ± 1.53	3 (2–5)	93.76	0.046
Fourth year	3.87 ± 1.29	4 (3–5)	109.68	

* Mann–Whitney U Test.

**Table 8 medicina-59-00116-t008:** Correlation between Score_Q4 and Score_Q11.

Correlation	*p* *
Score_Q4 (*p* < 0.001 **) × Score_Q11 (*p* < 0.001 **)	<0.001, R = 0.242

* Spearman’s rho Correlation Coefficient, ** Shapiro–Wilk Test.

## Data Availability

The data that support the findings of this study are available from the corresponding authors upon request.

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
