# Peer review of "Modular Digital and 3D-Printed Dental Models with Applicability in Dental Education"

_medicina, 2023, doi:10.3390/medicina59010116_

Round 1
Reviewer 1 Report
I have reviewed the manuscript, "Modular Digital and 3D-Printed Dental Models with Applicability in Dental Education" with focus on advancing a new method for creating modular digital dental models based on digitalization of standard dental models plus a strategy to obtain modular digital models based on real clinical cases.
The methodology presented by the authors is software-based and available to all licenced users so it would be worthwhile for the authors to explain the novelty or relevance of their study.
Yes, anyone with a 3D printer can produce a dental model from stl files - this is not new. "Evaluation of undergraduate students’ opinion on the use of modular digital and 3D-printed dental models in university dental education" can be a mere academic exercise when CAD/CAM is newly introduced in a curriculum. By asking leading questions in a questionnaire, the results will be predictable, so the survey adds little value to the study.
As an academic with expertise in CAD/CAM education, I'm afraid I cannot find anything new in this paper.
Author Response
Thank you for your recommendations and comments on our manuscript. We would like to show our gratitude for the useful remarks and constructive suggestions, which do help us significantly improve the quality of the current paper. All the comments and suggestions are appreciated.
We revised the manuscript according to the received comments.
The details of the revisions to the manuscript and our responses to the reviewers’ comments are written with blue coloured font in this cover letter - which is submitted as PDF File. Please see the attachment.
We also re-submit the new version of our manuscript, which has been revised, checked and modified after our careful analysis of the reviewers' comments.
Thank you for your support!
Kind regards,
Dr. Mihaela Pantea

Reviewer 2 Report
To authors:
The manuscript present important and interesting aspects regarding alternative digital proposal dedicated to dental teaching, within the field of prosthodontics. This precious manuscript can be glimpsed for a scientific publication within the scope of the Medicine, however the manuscript it seems possible publication some minimal questions should be considered by the authors.
I suggest splitting this manuscript into two distinct works. The first presenting technological/pedagogical innovation and the second its evaluation among undergraduate academic staff.
I congratulate and encourage the authors to finish this interesting manuscript with the requested suggestions to be corrected in the text. I suggest that the authors continue with this line of research.
General aspects:
The authors in their objectives suggest: (Objective 1): A new virtual technology allowing simulation of virtual cases on modular digital dental models and 3D-23 printing of the digital models obtained using a newly developed computerized method; (Objective 2): A questionnaire applied to undergraduate students in dentistry regarding this new pedagogical method.
We note that the manuscript is extensive because it contains two major important objectives compiled in a single study. My suggestion would be to split it into two works to be published in the same journal as part1 and part2. A descriptive work on this new pedagogical tool (1st part) and another work only with the evaluation of undergraduate students in dentistry (2nd part). This is just a “suggestion” and does not need to be accepted by the authors, just a suggestion to facilitate the reader's understanding of two important works (very well executed by the authors) that contain a connection between both.
1. In Materials and Methods
Pag. 8, line 213:
….can be 3D-printed. The STL files (Figure xx) were generated and ……
Which figure refers to xx described in this sentence?
2. In References
References need to be revised as several have a full formatting of newspapers and others abbreviated. Please check the publication rules and correct all of them in a standard way.
Take care with journals abbreviations, cursive formatting and/or missing data.
[See https://www.mdpi.com/authors/references]
Journal Articles:
Author 1, A.B.; Author 2, C.D. Title of the article. Abbreviated Journal Name Year, Volume, page range.
For example:
14. Schwendicke, F.A.; Samek, W.; Krois, J. Artificial intelligence in dentistry: chances and challenges. Journal of dental research. 2020, 99(7), 769-774.
17. Neville, P.; van der Zande, M.M. Dentistry, e-health and digitalization: A critical narrative review of the dental literature on digital technologies with insights from health and technology studies. Community dental health 2020, 37(1), 51-58.
21. Kato, A.; Ziegler, A.; Utsumi, M.; Ohno, K.; Takeichi, T. Three-dimensional imaging of internal tooth structures: applications in dental education. Journal of Oral Biosciences 2016, 58(3), 100-111.
31. Jalali P, Glickman GN, Umorin M. Do didactics improve clinical skills: A retrospective educational study. Saudi Endodontic Journal 2021, 11(1), 31. [missing page numbers]
32. Sjöström, M.; Brundin, M. The Effect of Extra Educational Elements on the Confidence of Undergraduate Dental Students Learning to Administer Local Anaesthesia. Dentistry Journal 2021, 9(7), 77.
35. Shaikh, S.; Nahar, P.; Ali, H.M. Current perspectives of 3D-printing in dental applications. Brazilian Dental Science 2021, 24(3). [missing page numbers]
39. Mahrous, A.; Schneider, G. B.; Holloway, J. A.; Dawson, D. V. Enhancing student learning in removable partial denture design by using virtual three-dimensional models versus traditional two-dimensional drawings: a comparative study. Journal of Prosthodontics 2019, 28(8), 927-933.
42. Richter, M.; Peter, T.; Rüttermann, S.; Sader, R.; Seifert, L. B. 3D printed versus commercial models in undergraduate conservative dentistry training. European Journal of Dental Education, 2022, 26(3), 643-651.
44. Soares, P. V.; de Almeida Milito, G.; Pereira, F. A.; Reis, B. R.; Soares, C. J.; de Sousa Menezes, M.; de Freitas Santos-Filho, P. C. Rapid prototyping and 3D-virtual models for operative dentistry education in Brazil. Journal of dental education 2013, 77(3), 358-363.
47. Shrivastava, K.J.; Nahar, R.; Parlani, S.; Murthy, V.J. A cross-sectional virtual survey to evaluate the outcome of online dental education system among undergraduate dental students across India amid COVID-19 pandemic. Eur J Dent Educ 2021. [missing page numbers]
52. Garcia, J.; Yang, Z.; Mongrain, R.; Leask, R.L.; Lachapelle, K. 3D-printing materials and their use in medical education: a review of current technology and trends for the future. BMJ Simulation and Technology Enhanced Learning 2018, 4, 27-40.
55. Ö zcan, M.; Hotza, D.; Fredel, M.C.; Cruz, A.; Volpato, C.A. Materials and Manufacturing Techniques for Polymeric and Ceramic Scaffolds Used in Implant Dentistry. Journal of Composites Science 2021, 5(3), 78.
63. Lee, K.Y.; Cho, J.W.; Chang, N.Y.; Chae, J.M.; Kang, K.H.; Kim, S.C.; Cho, J.H. Accuracy of three-dimensional printing for manufacturing replica teeth. The Korean Journal of Orthodontics 2015, 45(5), 217-25.
66. Kim, Y. K.; Kim, J. H.; Jeong, Y.; Yun, M. J.; Lee, H. Comparison of digital and conventional assessment methods for a single tooth preparation and educational satisfaction. European Journal of Dental Education, 2022.
67. Ishida, Y.; Kuwajima, Y.; Kobayashi, T.; Yonezawa, Y.; Asack, D.; Nagai, M.; Kondo, H.; Ishikawa-Nagai, S.; Da Silva, J.; Lee, S. J. Current Implementation of Digital Dentistry for Removable Prosthodontics in US Dental Schools. International Journal of Dentistry 2022.
68. Sharab, L.; Adel, M.; Abualsoud, R.; Hall, B.; Albaree, S.; de Leeuw, R.; Kutkut, A. Perception, awareness, and attitude toward digital dentistry among pre-dental students: an observational survey. Bulletin of the National Research Centre 2022, 46(1), 1-7.
71. Duane, B.; Dixon, J.; Ambibola, G.; Aldana, C.; Couglan, J.; Henao, D.; Daniela, T.; Nélio, Veiga, N.; Martin, N.; Darragh, J.H.; Ramasubbu, D.; Perez, F.; Schwendicke, F.; Correia, M.; Quinteros, M.; Van Harten, M.; Paganelli, PC.; Vos, P.; Lopez, R.M.; Field, J. Embedding environmental sustainability within the modern dental curriculum - Exploring current practice and developing a shared understanding. European Journal of Dental Education 2021, 25(3), 541-549.
73. Towers, A.; Field, J.; Stokes, C.; Maddock, S.; Martin, N. A scoping review of the use and application of virtual reality in pre-clinical dental education. Br Dent J 2019, 226(5), 358–366 (2019).
75. Chang, T.Y.; Hong, G.; Paganelli, C.; Phantumvanit, P.; Chang, W.J.; Shieh, Y.S.; Hsu, M.L. Innovation of dental education during COVID-19 pandemic. Journal of Dental Sciences 2021, 16(1), 15-20.
76. Kui, A.; Jiglau, A.L.; Chisnoiu, A.; Negucioiu, M.; Balhuc, S.; Constantiniuc, M.; Buduru, S. A survey on dental students’ perception regarding online learning during COVID-19 pandemic. Medicine and Pharmacy Reports 2022, 95(2), 203-208.
83. Moussa, R.; Alghazaly, A.; Althagafi, N.; Eshky, R.; Borzangy, S. Effectiveness of virtual reality and interactive simulators on dental education outcomes: systematic review. European Journal of Dentistry 2022, 16(01), 14-31.
Author Response
Dear Editor,
Thank you for your recommendations and comments on our manuscript. We would like to show our gratitude for the useful remarks and constructive suggestions, which do help us significantly improve the quality of the current paper. All the comments and suggestions are appreciated.
We revised the manuscript according to the received comments.
The details of the revisions to the manuscript and our responses to the reviewers’ comments are written with blue coloured font in this cover letter - which is submitted as PDF File. Please see the attachment.
We also re-submit the new version of our manuscript, which has been revised, checked and modified after our careful analysis of the reviewers' comments.
Thank you for your support!
Kind regards,
Dr. Mihaela Pantea

Round 2
Reviewer 2 Report
Authors:
I congratulate the authors for the corrections and suggestions adopted to finish this interesting manuscript. Considering the modifications, insertions and corrections adopted by authors, the manuscript is suitable for publication within the scope of Medicina.
I suggest that the authors continue with this line of research.